# Epitaxial Growth of Diamond-Shaped Au_1/2_Ag_1/2_CN Nanocrystals on Graphene

**DOI:** 10.3390/ma14247569

**Published:** 2021-12-09

**Authors:** Chunggeun Park, Jimin Ham, Yun Jung Heo, Won Chul Lee

**Affiliations:** 1Department of Mechanical Engineering, BK21 FOUR ERICA-ACE Center, Hanyang University, 55 Hanyangdaehak-ro, Sangnok-gu, Ansan 15588, Korea; cndrms5568@hanyang.ac.kr (C.P.); jiminham@hanyang.ac.kr (J.H.); 2Department of Mechanical Engineering and Integrated Education Institute for Frontier Science & Technology, Kyung Hee University, 1732 Deokyoungdae-ro, Giheung-gu, Yongin 17104, Korea

**Keywords:** metal cyanide, Au_1/2_Ag_1/2_CN, rhombic nanocrystal, van der Waals epitaxy, graphene, epitaxial alignment

## Abstract

Epitaxial synthesis of inorganic nanomaterials on pristine 2D materials is of interest in the development of nanostructured devices and nanocomposite materials, but is quite difficult because pristine surfaces of 2D materials are chemically inert. Previous studies found a few exceptions including AuCN, AgCN, CuCN, and Cu_0.5_Au_0.5_CN, which can be preferentially synthesized and epitaxially aligned onto various 2D materials. Here, we discover that Au_1/2_Ag_1/2_CN forms diamond-shaped nanocrystals epitaxially grown on pristine graphene surfaces. The nanocrystals synthesized by a simple drop-casting method are crystallographically aligned to lattice structures of the underlying graphene. Our experimental investigations on 3D structures and the synthesis conditions of the nanocrystals imply that the rhombic 2D geometries originate from different growth rates depending on orientations along and perpendicular to 1D molecular chains of Au_1/2_Ag_1/2_CN. We also perform in situ TEM observations showing that Au_1/2_Ag_1/2_CN nanocrystals are decomposed to Au and Ag alloy nanocrystals under electron beam irradiation. Our experimental results provide an additional example of 1D cyanide chain families that form ordered nanocrystals epitaxially aligned on 2D materials, and reveal basic physical characteristics of this rarely investigated nanomaterial.

## 1. Introduction

Epitaxial synthesis of inorganic nanomaterials on pristine surfaces of 2D materials [1,2,3,4,5] has attracted widespread interest in the fields of nanodevices and nanocomposite materials such as van der Waals heterostructures [1,2,3]. However, because pristine surfaces of 2D materials are chemically inert, it is difficult to assemble and align inorganic nanomaterials directly on to pristine 2D materials [4,5,6,7]. The majority of previous techniques are based on dangling bonds of 2D materials [6,8,9,10] and intermediate seed materials [11,12], thus only forming randomly oriented or poorly aligned inorganic nanostructures on 2D materials. Otherwise, vapor-phase deposition at high temperature [7,13,14,15,16] is required to synthesize ordered inorganic nanostructures directly on pristine 2D materials. Based on these previous studies, many researchers consider that such synthesis is quite difficult, even virtually impossible with aqueous-phase reaction under ambient conditions [4,5,6,7].

Metal cyanides are expected as an uncommon exception in the interactions between inorganic materials and pristine 2D materials, after it was initially discovered in 2015 [17] that AuCN can be epitaxially grown on graphene using aqueous-phase reaction at room temperature. Follow-up studies [17,18,19,20,21] revealed that AuCN, AgCN, CuCN, and Cu_0.5_Au_0.5_CN are epitaxially aligned onto various 2D materials including graphene, h-BN, and MoS_2_. One-dimensional (1D) molecular chain structures, which are a characteristic feature of these metal cyanides, are responsible for their preferential growth and epitaxial alignment on 2D materials, because the 1D chain structures would enable van der Waals epitaxy [22,23] and orientation-dependent interactions to hexagonal lattices [21]. These facts imply that the nature of preferential and epitaxial growth could be a universal characteristic of metal cyanides [21], which also have 1D chain structures. Although there exist various types of cyanide complexes based on single and mixed metal elements (Au, Ag, Cu, Hg, and Fe) [24,25,26,27], investigations on epitaxial growth have been performed only for the above-mentioned four materials (AuCN, AgCN, CuCN, and Cu_0.5_Au_0.5_CN). Therefore, a question is raised whether other types of metal cyanides form nanostructured materials epitaxially aligned on 2D materials.

In this study, we discover that Au_1/2_Ag_1/2_CN forms diamond-shaped nanocrystals epitaxially grown on pristine graphene surfaces (Figure 1). The nanocrystals synthesized by a simple drop-casting method are crystallographically aligned to lattice structures of the underlying graphene. This epitaxial alignment is likely to originate from the potential energy landscape of graphene’s hexagonal lattices similar to other metal cyanides, and our experimental investigations on 3D structures and synthesis conditions imply that the rhombic 2D geometries originate from different growth rates depending on orientations along and perpendicular to 1D Au_1/2_Ag_1/2_CN chains. We also perform in situ TEM observations of the decomposition process of Au_1/2_Ag_1/2_CN nanocrystals in which Au and Ag alloy nanocrystals are formed under electron beam irradiation. These results provide an additional example of 1D cyanide chain families that form ordered nanocrystals epitaxially aligned on 2D materials, and present basic material characteristics of Au_1/2_Ag_1/2_CN, which has been rarely investigated.

## 2. Materials and Methods

Sample preparation starts with transferring single-layered CVD graphene to TEM grids with carbon support films (lacey carbon film on gold mesh and Quantifoil holey carbon film on gold mesh) using a direct transfer method [28]. In brief, the lacey/holey carbon TEM grid is placed onto a graphene-covered copper foil with the carbon film side facing the graphene. Then 2.5 μL of isopropyl alcohol (IPA) is dropped onto the sample to wet the interface, and the sample is dried at 85 °C on a hot plate for 15 min to promote adhesion between the carbon film and the graphene. The grid is floated on an aqueous solution of 113 mM ammonium persulfate, (NH_4_)_2_S_2_O_8_, to etch the underlying copper foil and rinsed several times by floating the graphene transferred grid on deionized water. To prepare multi-layered graphene, graphite flakes are mechanically exfoliated on a 300 nm SiO_2_/Si wafer using adhesive tape. The synthesis of Au_1/2_Ag_1/2_CN nanocrystals using drop-casting (Figure 1a) starts from dissolving 1.6 mM AuCN and 1.6 mM AgCN respectively in aqueous solutions of ammonia (25~30%, SAMCHUN). One-to-one mixing of the two solutions generates 0.8 mM Au_1/2_Ag_1/2_CN solution, which is dropped on the prepared graphene substrate and is dried at 80 °C for 20 min. Transmission electron microscopy (TEM) observations are performed using a JEOL 2100F TEM operated at 200 kV.

## 3. Results and Discussion

### 3.1. Formation of Diamond-Shaped Au_1/2_Ag_1/2_CN Nanocrystals

Drop casting of 1:1 mixed aqueous solution of AuCN and AgCN (Figure 1a) forms diamond-shaped (rhombic) nanocrystals on graphene, as shown in typical TEM images in Figure 1c,e. While the synthesized nanocrystals have a wide range of sizes from ~10 nm to ~1 μm, 2D projected images of the nanocrystals have a uniform shape of a sharp rhombus (diamond). High-resolution TEM images in Figure 1e,f shows that the nanocrystals have a regular lattice pattern with a spacing of 5.2 Å. The orientation of lattice lines is parallel to a longer diagonal of the nanocrystal rhombus. A selected area electron diffraction (SAED) pattern in Figure 1d and a fast Fourier transformed (FFT) image in Figure 1g also confirm that the diamond-shaped nanocrystals have lattice patterns with a spacing of 5.15 ± 0.05 Å, which coincides well with a previously-known lattice spacing (5.158 Å) of Au_1/2_Ag_1/2_CN [24]. We measure a Raman spectrum of the diamond-shaped nanocrystals synthesized on multi-layered graphene (Figure 2a). The measured Raman spectrum presents a principal *ν*_C≡N_ vibration near 2225 cm^−1^ in agreement with the previous work about Au_1/2_Ag_1/2_CN [24].

The above observations indicate that Au_1/2_Ag_1/2_CN forms diamond-shaped nanocrystals on graphene. The possibility that the observed diamond-shaped nanocrystals are AuCN or AgCN is very low because of the following three reasons. First, the measured lattice spacing (5.15 ± 0.05 Å) differs from the major lattice spacings (5.07 Å or 3.60 Å) of AuCN or AgCN [24,27]. Second, the rhombic geometry seems unique because both AuCN and AgCN form horizontally grown nano/micro-sized wires on 2D material surfaces including graphene [17,18,19,20,21,29]. Third, energy-dispersive X-ray spectroscopy (EDX) in Appendix A shows both gold and silver elements from the synthesized nanocrystals. In addition, the possibility that the observed nanocrystals are made out of Au_x_Ag_(1−x)_CN is also very low due to 1:1 stoichiometry in the precipitation of AuCN and AgCN mixture. Previous attempts to make gold-rich or silver-rich compositions [24] resulted in the formation of the Au_1/2_Ag_1/2_CN solid phase, and excessive amounts of gold or silver formed AuCN or AgCN. Au_1/2_Ag_1/2_CN is known to form 1D chains of [Au–C≡N–Ag–N≡C–]_n_ with complete ordering, and the chains are stacked by van der Waals interactions to form hexagonal crystals [24]. In the synthesized Au_1/2_Ag_1/2_CN nanocrystals, the orientation of 1D chains is parallel to a shorter diagonal of the nanocrystal rhombus. Aligned Au and Ag atoms make lattice lines perpendicular to the 1D chain direction, which are parallel to a longer diagonal of the rhombus.

### 3.2. Epitaxial Alignment between Au_1/2_Ag_1/2_CN Nanocrystals and Graphene

One of the most interesting points in the Au_1/2_Ag_1/2_CN nanocrystals is that the nanocrystals have preferential growth directions that are related to the underlying graphene lattice structures. In the left area of Figure 1c, all diamond-shaped nanocrystals with various sizes are preferentially aligned to a specific direction, and the largest nanocrystal near the center of Figure 1c is oriented to the 120°-rotated direction. The SAED pattern in Figure 1d clearly shows the epitaxial alignment between the nanocrystals and graphene. The diffraction peaks from the Au_1/2_Ag_1/2_CN nanocrystals (circled in yellow) show good orientational alignment to the second-order diffraction peaks—(1–210) peaks—of graphene (circled in red), which means that 1D chains of Au_1/2_Ag_1/2_CN are preferentially aligned to the zigzag lattice directions of the underlying graphene in real lattice space. Because the 1D chains of Au_1/2_Ag_1/2_CN are parallel to a shorter diagonal of the nanocrystal rhombus, longer diagonals of the nanocrystal rhombuses are aligned to the armchair lattice directions of graphene.

The above observations raise two questions: the first is how Au_1/2_Ag_1/2_CN can grow preferentially on chemically inert surfaces of graphene, and the second is what makes the alignment between the two materials. The first question can be answered with van der Waals epitaxy [22,23], which is the heteroepitaxial relationship between two material surfaces having no dangling bonds. The atoms in a Au_1/2_Ag_1/2_CN chain are bound to each other with strong covalent bonds, whereas the chains are held together via van der Waals forces [24]. Therefore, the Au_1/2_Ag_1/2_CN crystal can be easily cleaved along the chains without producing any dangling bonds on its surface. Because this surface can generate van der Waals epitaxial interaction with pristine graphene surfaces, Au_1/2_Ag_1/2_CN nanocrystals can grow preferentially on graphene. It is noteworthy that in van der Waals epitaxy [22,23], lattice mismatching is maintained (not relaxed) even at the interface between Au_1/2_Ag_1/2_CN and graphene (Appendix A). The second question has been investigated in previous studies, which experimentally showed the alignment of 1D cyanide chains of AuCN, AgCN, CuCN, and Cu_0.5_Ag_0.5_CN along the zigzag lattice directions of various 2D materials including graphene, h-BN, WS_2_, MoS_2_, MoTe_2_, and WTe_2_ [17,18,19,20,21]. They explained that this epitaxial alignment originates from the potential energy landscape of the hexagonal lattices of 2D materials [21]. In detail, energetically favorable locations for metal cyanide adsorption onto graphene are hexagonally arranged, and, because of this hexagonal arrangement, energetically favorable orientations of a straight molecular chain are along zigzag lattice directions of graphene [21]. This mechanism can also explain the crystallographic alignment between Au_1/2_Ag_1/2_CN and graphene in our study. We would note that the strong interaction of inorganic nanomaterials on pristine surfaces of 2D material surfaces (without dangling bonds) can be uncommon [4,5,6,7], and our study provides an additional example of 1D cyanide chain families that shows the epitaxial alignment on pristine graphitic surfaces.

### 3.3. Geometrical Characteristics of the Au_1/2_Ag_1/2_CN Nanocrystals

We further investigate the geometrical characteristics of the Au_1/2_Ag_1/2_CN nanocrystals synthesized on graphene. Figure 2b shows interior angles of nanocrystal rhombuses (2D projected images of nanocrystals) measured from the synthesized nanocrystals in Figure 1c. As described in the inset of Figure 2b, we measure acute angles at the vertices whose connecting line is the longer diagonal of rhombuses. The histogram indicates that the averaged angle and its standard deviation are 47.4° and 6.8°, respectively. The synthesized nanocrystals have sharp acute angles, which are possible but uncommon for nanometer-sized crystals because obtuse angles are generally preferred in nanocrystals to minimize their surface energies. To investigate three-dimensional (3D) structures of the Au_1/2_Ag_1/2_CN nanocrystals, we capture scanning electron microscopy (SEM) images without and with sample tilting (Figure 2c,d; SEM images of additional samples are presented in Appendix A). Interestingly, the nanocrystal geometries do not follow obvious expectations such as thin rhombic plates or rhombic pyramids. Longer diagonals of nanocrystal rhombuses in 2D projected images are half circles in the vertical cross-sections of 3D nanocrystal structures. As shown in Figure 3e, the nanocrystal geometries can be described as a 3D structure made by cutting a circular cone in half vertically and then attaching two semicircular faces of the two identical pieces to each other. This 3D structure coincides well with contrast differences in TEM images. In TEM images of individual Au_1/2_Ag_1/2_CN nanocrystals (such as Figure 1e), the longer diagonal of the nanocrystal rhombus shows high contrast (relatively dark) near the center while the image near the outside edges shows low contrast (relatively bright gray).

One of the notable geometrical characteristics of the Au_1/2_Ag_1/2_CN nanocrystals is the sharp rhombic shape of their 2D projected images, and we check whether this characteristic is maintained in Au_1/2_Ag_1/2_CN nanocrystals synthesized under different conditions, including Au_1/2_Ag_1/2_CN concentrations in aqueous solutions and drying temperatures. Figure 3 shows TEM images of the nanocrystals synthesized by drying 0.8 mM Au_1/2_Ag_1/2_CN solution at 80 °C (Figure 3a,d,g; an identical condition to synthesize the sample in Figure 1c), by drying 0.8 mM Au_1/2_Ag_1/2_CN solution at 150 °C (Figure 3b,e,h), and by drying 0.08 mM Au_1/2_Ag_1/2_CN solution at 80 °C (Figure 3c,f,i). From these TEM images, we obtain histograms of interior angles of the rhombic Au_1/2_Ag_1/2_CN nanocrystals synthesized with the three different conditions (Figure 3g–i). The first case (Figure 3a,d,g) whose synthesis condition (0.8 mM and 80 °C) is identical to that of Figure 1 shows sharp rhombic nanocrystals whose averaged interior angle is 52.5 ± 7.3°. This value is close to the averaged interior angle measured from Figure 1c (47.4 ± 6.8° in Figure 2b) if we consider standard deviations, while Figure 1c and Figure 3a have different nanocrystal sizes and densities. We would note that, because the synthesis method in this study is drop-casting, it is very hard to stably control the nanocrystal sizes and densities: the sizes and densities vary largely even in a single sample. We would also note that nanowires with sharp tips in Figure 3a are not epitaxially-grown nanocrystals, but nanocrystals formed in bulk solution. The above comparison between Figure 1c and Figure 3a shows that the major geometrical characteristic of Au_1/2_Ag_1/2_CN nanocrystals (interior angle) is maintained in different samples synthesized under an identical condition. The second case (Figure 3b,e,h) using a higher drying temperature (150 °C) and an identical concentration (0.8 mM) also shows rhombic nanocrystals but their interior angles are increased to 67.1° ± 7.5°. Changes in nanocrystal geometries are clearer in the third case (Figure 3c,f,i) using a lower concentration (0.08 mM) and the identical drying temperature (80 °C). In Figure 3c,f, 2D projected images of the Au_1/2_Ag_1/2_CN nanocrystals show square-like shapes, and their interior angles are largely increased to 92.5 ± 9.1° (close to a right angle). Although additional experiments are needed to reveal detailed relationships between synthesis conditions and nanocrystal geometries, Figure 3 clearly shows that crystallographic orientations forming shape edges in the nanocrystals are changed depending on synthesis conditions.

The observed dependence of the nanocrystal shapes on the synthesis conditions would be strongly related to their formation mechanism. In metal nanocrystals, one of the important mechanisms to form polyhedral structures is the development of low-energy facets during the growth process [30]. This is not the case in our study because a specific crystallographic facet is not developed in the diamond-shaped nanocrystals. The formation of rhombic 2D geometries seems to originate from different growth rates depending on orientations along and perpendicular to 1D molecular chains of Au_1/2_Ag_1/2_CN during the precipitation process. In the original synthesis condition (samples in Figure 1 and Figure 3a), the axial growth rate of 1D Au_1/2_Ag_1/2_CN chains might be slower than the growth rate by stacking 1D Au_1/2_Ag_1/2_CN chains, thus forming the sharp rhombic geometry whose shorter diagonal is parallel to the 1D chain direction. In the different synthesis conditions (samples in Figure 3a,b), the difference in the growth rates might be reduced, thus the rhombic 2D geometry becomes closer to squares. This growth model also coincides well with 3D structures of the nanocrystals observed with SEM (Figure 2c–e and Appendix A). Because the stacking speed of 1D Au_1/2_Ag_1/2_CN chains determines growth rates perpendicular to the chain’s axial direction, all growth rates along these perpendicular directions in 3D space would be identical except the directions blocked by graphene. Therefore, cross-sections perpendicular to 1D chain directions are half circles in the synthesized nanocrystals. The above hypothesis can successfully explain the observations in this study, but additional investigations are required to clearly determine the formation mechanism of rhombic Au_1/2_Ag_1/2_CN nanocrystals.

### 3.4. In Situ TEM Observation of e-Beam-Induced Decomposition of Au_1/2_Ag_1/2_CN

During the TEM imaging of the Au_1/2_Ag_1/2_CN nanocrystals, we can discover material transformations under electron beam irradiation. A typical in situ TEM observation (acceleration voltage: 200 kV, electron dose-rate: ~5000 e^−^/Å^2^s) is presented in Appendix A with snapshots in Figure 4a, showing that the electron beam induces the solid-phase transformation from Au_1/2_Ag_1/2_CN to metal nanocrystals. The beginning of the movie shows a crystalline area that has a lattice spacing of 5.16 Å, which is a characteristic spacing of Au_1/2_Ag_1/2_CN. This lattice spacing is clearly larger than the typical lattice spacings of Au or Ag FCC (face cubic centered) crystals, which are ~2.3 Å, ~2.0 Å, and ~1.4 Å from (111), (200), and (220) planes. Therefore, Au_1/2_Ag_1/2_CN can be easily identified with the lattice spacing of 5.16 Å in high-resolution TEM observations. The Au_1/2_Ag_1/2_CN crystal in the first frame of the movie is decomposed during the TEM observation. The snapshot at 15 min contains no lattice spacing of 5.16Å, and Au_1/2_Ag_1/2_CN peak intensities over time in FFT images (Figure 4b) of TEM movie frames also confirm the Au_1/2_Ag_1/2_CN decomposition as shown in Figure 4c. During this in situ TEM observation, metallic nanocrystals (black dots in Figure 4a) are formed near the Au_1/2_Ag_1/2_CN nanocrystal and grow (up to ~10 nm) by atomic attachment as well as nanoparticle coalescence. The metallic nanocrystals have the above-mentioned lattice spacings (~2.3 Å, ~2.0 Å, and ~1.4 Å), and would be bimetallic FCC nanocrystals composed of Au and Ag. The lattice patterns of bimetallic nanocrystals are clearly shown in Appendix A with enlarged snapshots in Figure 4d, in which the processes of Au_1/2_Ag_1/2_CN decomposition and bimetallic nanocrystal growth are observed with in situ TEM. The lattice patterns of bimetallic nanocrystals are randomly orientated, and do not show any epitaxial alignment along the underlying graphene unlike their precursor material (Au_1/2_Ag_1/2_CN). It has been shown that metal cyanide complexes can be easily decomposed in the solid phase because the ionic bonding between metal cations and CN^−^ ions is relatively weak [31]. These facts were experimentally shown in the 1990s from ex situ studies of solid-phase thermolysis of several metal cyanides to metallic Au, Ag, Cu, Hg, and Fe (at ~300–~400 °C) [25,26,32]. Recent studies have shown that gold(I) cyanide (AuCN) can be decomposed to form metallic gold by various energy sources including heat, electron beams, ion beams, and optical laser [17,31,33]. In addition, the formation process of Au nanocrystals from AuCN has been observed in real time with high-resolution in situ TEM [17,29]. Because the electron beam decomposes AuCN to zerovalent Au atoms and gaseous (CN)_2_, the Au atoms hold together to form Au nanocrystals [29,31]. Based on these previous studies, the Au_1/2_Ag_1/2_CN decomposition process discovered in this study can be described as follows: [Au–CN–Ag–NC–]_n_ + electron beam → n(AuAg) + n(CN)_2_↑

The electron beam for TEM imaging acts as a reducing agent to decompose the Au_1/2_Ag_1/2_CN nanocrystal (radiolysis) to zerovalent Au or Ag atoms and (CN)_2_ gas. Then, the densities of Au and Ag atoms increase locally near the imaged Au_1/2_Ag_1/2_CN nanocrystal, and the reduced metal atoms aggregate to form AuAg alloyed nanocrystals. We envision that this newly-discovered process can be a good model system to investigate atomistic mechanisms of the early-stage process of metallic alloy formation, as in situ TEM observations of the AuCN decomposition process to Au nanocrystals delivered mechanistic understandings of crystal nucleation at the atomic level [29].

## 4. Conclusions

In this study, we have presented the epitaxial growth of diamond-shaped Au_1/2_Ag_1/2_CN nanocrystals on pristine graphene surfaces. The nanocrystals synthesized through simple drop-casting are crystallographically aligned to lattice structures of the underlying graphene. This epitaxial alignment would originate from the potential energy landscape of graphene’s hexagonal lattices similar to other metal cyanides, and experimental investigations on 3D structures and synthesis conditions imply that the rhombic 2D geometries originate from different growth rates depending on orientations along and perpendicular to 1D Au_1/2_Ag_1/2_CN chains. In addition, in situ TEM observations show the decomposition process of Au_1/2_Ag_1/2_CN nanocrystals to Au and Ag alloy nanocrystals under electron beam irradiation. These results provide an additional example of 1D cyanide chain families that forms ordered nanocrystals epitaxially aligned on 2D materials and reveal basic physical characteristics of the rarely investigated nanomaterial, Au_1/2_Ag_1/2_CN.

## Figures and Tables

**Figure 1 materials-14-07569-f001:**
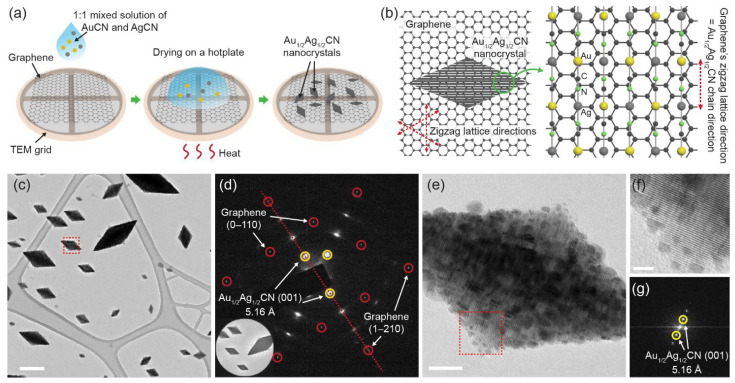
Diamond-shaped Au_1/2_Ag_1/2_CN nanocrystals epitaxially grown on single-layered graphene: (**a**) Nanocrystal synthesis method based on drop-casting. The sample from which (**c**–**g**) are obtained is prepared by dropping 0.8 mM Au_1/2_Ag_1/2_CN aqueous solution onto CVD-synthesized graphene and drying it at 80 °C; (**b**) schematic view showing the epitaxial alignment between Au_1/2_Ag_1/2_CN and graphene. Lattice spacings of the two materials are presented in Appendix A; (**c**) TEM image of the diamond-shaped Au_1/2_Ag_1/2_CN nanocrystals synthesized on graphene. Scale bar: 200 nm; (**d**) selected area electron diffraction (SAED) pattern of the nanocrystal-graphene sample. The inset shows an area in (**c**) where the SAED pattern is measured; (**e**) TEM image of an individual Au_1/2_Ag_1/2_CN nanocrystal (enlarged view of box in (**c**)). Scale bar: 20 nm; (**f**) enlarged view of box in (**e**). Scale bar: 5 nm; (**g**) fast Fourier transformed (FFT) image of (**e**).

**Figure 2 materials-14-07569-f002:**
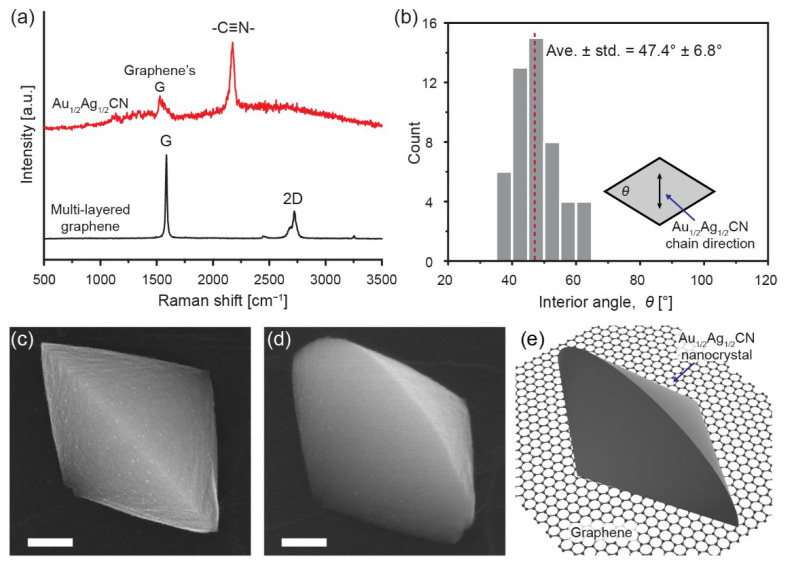
Geometrical and optical characteristics of the diamond-shaped Au_1/2_Ag_1/2_CN nanocrystals: (**a**) Raman spectrum of the Au_1/2_Ag_1/2_CN nanocrystals on multi-layered graphene; (**b**) histogram of interior angles of the Au_1/2_Ag_1/2_CN nanocrystals in Figure 1c. The inset indicates the measured interior angle in an Au_1/2_Ag_1/2_CN rhombus (diamond); (**c**,**d**) SEM images of an Au_1/2_Ag_1/2_CN nanocrystal from the top (**c**) and with tilting of 40° (**d**). Scale bars: 200 nm; (**e**) schematic view of the Au_1/2_Ag_1/2_CN nanocrystal, whose 3D structure can be made by cutting a circular cone in half vertically and then attaching two semicircular faces of the two half-cones.

**Figure 3 materials-14-07569-f003:**
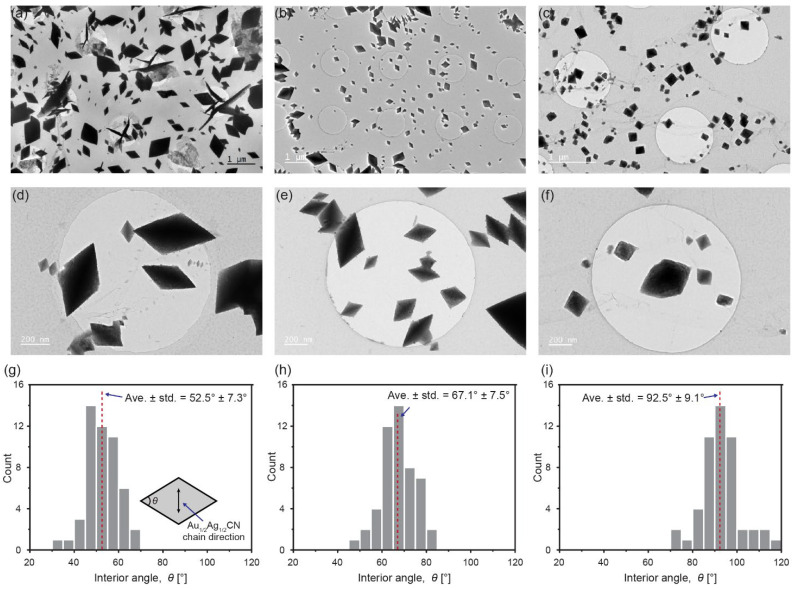
Diamond-shaped Au_1/2_Ag_1/2_CN nanocrystals synthesized under different conditions (Au_1/2_Ag_1/2_CN concentrations and drying temperatures): (**a**,**d**,**g**) 0.8 mM and 80 °C; (**b**,**e**,**h**) 0.8 mM and 150 °C; (**c**,**f**,**i**) 0.08 mM and 80 °C; (**a**–**f**) TEM images of the synthesized Au_1/2_Ag_1/2_CN nanocrystals synthesized on single-layered graphene; (**g**–**i**) histograms of interior angles of the Au_1/2_Ag_1/2_CN nanocrystals in a–f. The inset in g indicates the measured interior angle in an Au_1/2_Ag_1/2_CN rhombus (diamond).

**Figure 4 materials-14-07569-f004:**
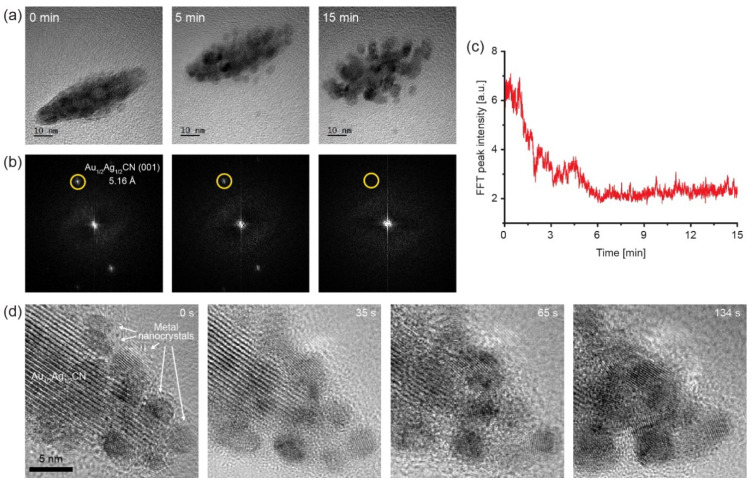
Decomposition process of the Au_1/2_Ag_1/2_CN nanocrystal to Au/Ag alloy nanoparticles under electron-beam irradiation: (**a**) A series of TEM images of the Au_1/2_Ag_1/2_CN nanocrystal from Appendix A; (**b**) Fourier transformed images of the TEM images in (**a**); (**c**) intensity of the Au_1/2_Ag_1/2_CN (001) peaks (yellow-circled dots) in (**b**) over time; (**d**) a series of TEM images from Appendix A. Lattice patterns of growing metal nanocrystals can be observed during the in situ TEM imaging.

## Data Availability

The data presented in this study are available in the article and Appendix A.

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
