# Peer review of "Epitaxial Growth of Diamond-Shaped Au1/2Ag1/2CN Nanocrystals on Graphene"

_materials, 2021, doi:10.3390/ma14247569_

Round 1
Reviewer 1 Report
The authors present an experimental study of the growth of 3D AuAg cyanide nanocrystals on graphene by drop casting at elevated temperature. The structure of the rhombic deposits is investigated by transmission electron microscopy (TEM) and atomic force microscopy (AFM). By statistical analysis, a preferential epitaxial alignment of the nanocrystals with respect to the underlying honeycomb lattice is identified. Furthermore, the decomposition of the cyanide nanocrystals into metallic AuAg nanocrystals upon electron irradiation is studied in situ.
Summarizing, the authors present interesting results that, in principle, might merit publication in “Materials”. However, in the present state several parts of the analysis still appear somewhat preliminary and should be expanded. Also, the physical insights that are conveyed to the reader appear quite limited. In terms of presentation and style, the paper is well written in general; yet, there are a number of deficiencies that also need to be addressed upon revision. Below, I am giving a list of suggestions that may serve as a basis for how to improve the overall scientific impact and its readability.
(1) The introduction section is quite shallow, partially redundant or even circular (e.g., lines 28 to 35). Possible additions could be explanations as to why the cyanides may be an exception to the rule of weak chemical interaction with the layered materials, the issue of the “dangling bonds”, the importance of defects, etc.
(2) It remains unclear how the atomic geometry presented in Figure 1b is determined and quantified. What is the evidence (experimental or theoretical) for this specific registry?
(3) Section 3.1: The chemical composition should be quantified from the TEM-EDX analysis.
(4) The argument based on the potential energy landscape as the origin of the epitaxial alignment is very generic and should, if possible, be extended in terms of plausibility.
(5) Figures 2b, 3g-i: The statistics presented need to be improved considerably to strengthen the result. The analysis of 20 islands is insufficient given the spread in angles measured.
(6) Figure 2c: An AFM image should be presented to assert the quality of the data as measuring such large height differences may be experimentally challenging.
(7) The issue of 3D nanocrystal shape should ideally be investigated by scanning electron microscopy show the real-space sample morphology at tilted angle of incidence.
(8) The dependence of the nanocrystal shape and density on the synthesis conditions is intriguing and would certainly merit further investigation. Specifically, how can such a change in nanocrystal shape from rhombic to cubic be explained? XRD may be of help here.
(9) The observation of the decomposition process of the nanocrystals is interesting, but provides only limited physical insights into the underlying mechanisms. This part of the paper may also be expanded.
Reviewer 2 Report
This manuscript presents a detailed TEM study of Au1/2Ag1/2CN nanocrystals grown on graphene. The synthesis of the nanocrystals and the study are in line with previous works by the same authors with different metal cyanide compositions. The results are different enough to make this work original compared to the previously published papers. The manuscript is sound, clearly written and easy to follow. For these reasons I recommend to accept this manuscript in Materials. Below are some minor corrections and comments which the author should consider before publication.
1/ page 5, line 193: the concentration, temperature and figure references are identical for the first and third synthesis conditions. According to line 210 page 6, the concentration for the third synthesis condition should be 0.08 mM (instead of 0.8 mM) and the figures 3c, 3f and 3i (instead of 3a, 3d and 3g).
2/ page 6, line 240: the lattice spacings for FCC AuAg nanocrystals are given but the corresponding diffraction data are not shown. I guess the epitaxial growth on graphene is not conserved after decomposition but the authors should consider addressing this point in the manuscript.
3/ Figure 2a: the graphene G band in the presence of the nanocrystals is shifted toward lower Raman shifts and broader compared to pristine graphene. An explanation or interpretation would be welcomed.
